# Changes in Energy Status of *Saccharomyces cerevisiae* Cells during Dehydration and Rehydration

**DOI:** 10.3390/microorganisms9020444

**Published:** 2021-02-21

**Authors:** Neringa Kuliešienė, Rasa Žūkienė, Galina Khroustalyova, Chuang-Rung Chang, Alexander Rapoport, Rimantas Daugelavičius

**Affiliations:** 1Department of Biochemistry, Faculty of Natural Sciences, Vytautas Magnus University, LT-44404 Kaunas, Lithuania; neringa.kuliesiene@vdu.lt (N.K.); rasa.zukiene@vdu.lt (R.Ž.); 2Institute of Microbiology and Biotechnology, University of Latvia, Jelgavas Str., 1-537, LV-1004 Riga, Latvia; galinah@lanet.lv (G.K.); rapoport@mail.eunet.lv (A.R.); 3Institute of Biotechnology, National Tsing Hua University, Hsinchu City 30013, Taiwan; crchang@life.nthu.edu.tw

**Keywords:** yeast, anhydrobiosis, dehydration–rehydration, metabolism, mitochondria

## Abstract

Anhydrobiosis is the state of life when cells are exposed to waterless conditions and gradually cease their metabolism. In this study, we determined the sequence of events in *Saccharomyces cerevisiae* energy metabolism during processes of dehydration and rehydration. The intensities of respiration and acidification of the medium, the amounts of phenyldicarbaundecaborane (PCB^−^) bound to yeast membranes, and the capabilities of cells to accumulate K^+^ were assayed using an electrochemical monitoring system, and the intracellular content of ATP was measured using a bioluminescence assay. Mesophilic, semi-resistant to desiccation *S. cerevisiae* strain 14 and thermotolerant, very resistant to desiccation *S. cerevisiae* strain 77 cells were compared. After 22 h of drying, it was possible to restore the respiration activity of very resistant to desiccation strain 77 cells, especially when glucose was available. PCB^−^ binding also indicated considerably higher metabolic activity of dehydrated *S. cerevisiae* strain 77 cells. Electrochemical K^+^ content and medium acidification assays indicated that permeabilization of the plasma membrane in cells of both strains started almost simultaneously, after 8–10 h of desiccation, but semi-resistant strain 14 cells maintained the K^+^ gradient for longer and more strongly acidified the medium. For both cells, the fast rehydration in water was less efficient compared to reactivation in the growth medium, indicating the need for nutrients for the recovery. Higher viability of strain 77 cells after rehydration could be due to the higher stability of their mitochondria.

## 1. Introduction

Anhydrobiosis is one of the special forms of cryptobiosis, in which the yeasts survive through adverse environmental conditions, in this case, the absence of water (desiccation). To survive, cells adapt to the changed environment, temporarily suspending their metabolism [1,2,3]. Understanding of the mechanism of anhydrobiosis and the main determinants important for cell viability after transfer to the state of anhydrobiosis and knowledge of events during the subsequent processes of rehydration and reactivation are important for fundamental science and could be applied in biomedicine and biotechnology. In biomedicine, an understanding of anhydrobiosis is important for research in translational medicine and the production of new vaccines. Further improvement of the quality of dry biopreparations is needed in biotechnology, including active dry yeasts for wine, beer, and ethanol production, as well as for the production of various biologically active compounds [4,5]. Big companies provide high-quality preparations of dry baker’s yeast with a very high level of cell viability. At the same time, such a result still cannot be achieved for other yeast species, as well as for other *S. cerevisiae* strains necessary for practical goals. One of the approaches to understand the reasons for these differences is the comparison of yeasts demonstrating various resistance patterns during dehydration–rehydration treatment. Previous studies showed [6], that osmotolerant halophilic yeasts *Debaryomyces hansenii* are more resistant to dehydration compared to *S. cerevisiae*. Being in the exponential phase of growth, these cells can effectively transit into anhydrobiosis, when other actively growing yeast species are very sensitive to such treatment. Further research revealed essential differences in the chemical composition of their plasma membranes, which led to a lower phase transition temperature of the membrane lipids [7,8]. In another study, it was revealed that thermotolerant strains of S. cerevisiae are more resistant to dehydration compared to mesophilic strains [8]. Recently, it was shown [9] that psychrophilic yeast species are also more resistant to dehydration than mesophilic ones. These results mean that yeast strains and species able to survive in more extreme environmental conditions also possess higher resistance to dehydration–rehydration treatment. Therefore, we expected that continuation of the studies directed at the comparison of strains with different resistances to dehydration–rehydration treatment may lead to information that is important for general biology, as well as facilitate the development of technologies for the improvement of the resistance of sensitive strains, which is important for current biotechnology.

The purpose of this study was to compare mesophilic and thermotolerant strains of *S. cerevisiae* to determine the sequence of events in the energy metabolism of these cells during dehydration and the regeneration of these processes during rehydration. The intensities of respiration and acidification of the medium, the amount of phenyldicarbaundecaborane (PCB^−^) bound to cellular membranes, and the capability of cells to accumulate K^+^ were assayed using an electrochemical monitoring system. The results of our experiments indicated that very resistant to desiccation strain 77 cells show respiration activity after 22 h of drying, demonstrate a stronger plasma membrane barrier to lipophilic anions, and maintain a higher content of intracellular ATP, although mesophilic strain 14 cells maintained the K^+^ gradient for longer during desiccation and more strongly acidified the incubation medium. It was concluded that the higher viability of rehydrated strain 77 cells could be due to the higher stability of their mitochondria. These results advance our understanding of the impact of dehydration on energy metabolism in yeasts.

## 2. Materials and Methods

### 2.1. Strains and the Cultivation Conditions

The diploid isogenic yeast strains used in this study, *S. cerevisiae* 14 and *S. cerevisiae* 77, were obtained from the Microbial Strain Collection of Latvia (http://mikro.daba.lv, accessed on 4 February 2021). The cells were grown in conical flasks, 1/5 of the volume filled with YPD medium (1% yeast extract (Acros Organics), 2% peptone (Oxoid), and 2% glucose (Chempur), in a thermostatted shaker at 30 °C for 44 h of shaking at 220 rpm (to the middle of the stationary growth phase).

### 2.2. Preparation of Cells for Experiments and Desiccation

For dehydration experiments, the cells were grown as described above, harvested by centrifugation (1000× *g*, 10 min), washed twice with 0.1 M sodium phosphate buffer, pH 7, then resuspended in 5 mL of the same buffer. After measuring OD_600_ of the diluted suspension, the exact volumes of the suspensions, containing 1.7 × 10^8^ cells (in the conditions used, OD_600_ 1 corresponded to 3 × 10^7^ cells/mL), were aliquoted into 1.5 mL Eppendorf-type tubes. The cells were pelleted by centrifugation (1000× *g*, 3 min) and supernatants were carefully removed. Open tubes with the cell pellets were placed into an oven and desiccated at 30 °C for 22 h. After drying, open tubes with the dry yeast pellets were kept in an exicator for 1–3 days, in the presence of silica gel. For the control of 0% humidity, the cells were incubated at 105 °C for 24 h.

### 2.3. Cell Rehydration and Reactivation

Fifty microliters of deionized water (for rehydration) or the same volume of YPD growth medium (for reactivation) were poured into the tubes with the dry yeast pellet, and the suspension was vortexed and incubated at room temperature for defined periods of time. The contents of two tubes were transferred into one vessel for electrochemical measurements. For K^+^ measurement, to determine the amount of this ion inside the cells, after incubation in YPD, the cells were washed twice with 0.1 M sodium phosphate buffer, pH 7.

### 2.4. Electrochemical Measurements

Several parameters of the metabolic activity of yeast cells were investigated: changes in the medium concentration of dissolved oxygen (respiration rate), ability of the cells to acidify the incubation medium (intensity of glycolysis), K^+^ and phenyldicarbaundecaborane (PCB^−^) concentrations in the incubation medium (cells’ capability to accumulate K^+^ and to maintain the metabolism-dependent plasma membrane barrier to lipophilic anions). The intensities of respiration and acidification of the medium, the level of PCB^−^ binding, and the amount of accumulated K^+^ were assayed using an electrochemical monitoring system [10]. Selective electrodes for K^+^ (Thermo Orion, model 9300BN), H^+^ (Hanna Instruments, model HI1131B), and PCB^−^ (prepared as described in [10]) and the dissolved oxygen probe (Thermo Orion, model 9708) were connected to an electrode potential-amplifying system with an ultralow-input bias current operational amplifier AD549JH (Analog Devices, Norwood, MA, USA). The data acquisition system PowerLab 8/35 (ADInstruments, Oxford, UK) was used to connect the amplifying system to a computer. Agar salt bridges were used for indirect connection of the Ag/AgCl reference electrodes (Orion model 9001; Thermo Fisher Scientific) with cell suspensions in the vessels. Contents of two Eppendorf tubes were transferred into thermostatted and magnetically stirred vessels and filled with 0.1 M sodium phosphate, pH 7, up to 5 mL. For acidification studies, the dry cells were resuspended and vessels were filled with medium containing 95 mM NaCl and 5 mM sodium phosphate, pH 7. Calibrations of K^+^- and PCB^−^-selective electrodes were performed before the addition of cells. During measurements of dissolved oxygen and H^+^ concentration in the cell suspensions, calibrations were performed at the end of the experiment, adding solid Na_2_S_2_O_4_ up to ~20 mM (0% dissolved oxygen, when 100% is the concentration in the medium before the addition of cells) or 10 µL of 0.1 M HCl, respectively. Up to 0.8% glucose was added to energize the cells and up to 10 μg/mL nystatin to achieve complete permeabilization of the cell plasma membrane. Measurements were performed at 30 °C.

### 2.5. ATP Measurements

The intracellular amounts of ATP were determined using a Modulus luminometer (Turner BioSystems, model 9200-003) and luciferin–luciferase method, taking samples for analysis directly from the vessels for electrochemical measurements [10] after 10 min of incubation in 0.1 M sodium phosphate buffer, pH 7. An ATP Biomass Kit (BioThema) was used in these experiments. Five microliters of the cell suspensions from the vessels were mixed with 5 μL of the lysing reagent, and after 10 min, 90 μL of a luciferin–luciferase mixture were added. The recorded intensity of luminescence was evaluated using an ATP calibration curve.

## 3. Results

### 3.1. Studies of Dehydration

Intensities of the respiration and acidification of the medium, the binding of PCB^−^ to cellular membranes, and the amount of cell-accumulated K^+^ were determined using an electrochemical monitoring system. Cells of two *S. cerevisiae* strains were compared: mesophilic, semi-resistant to desiccation strain 14 and thermotolerant, very resistant to desiccation strain 77. Before dehydration, strain 14 cells demonstrated higher energy status than strain 77 ones; the intensity of the respiration of strain 14 cells was higher (Figure 1A,B) and they accumulated more K^+^ (Figure 1C,D). The addition of glucose to the incubation medium induced a very strong decrease in the dissolved oxygen concentration in suspensions of both cells, but this effect gradually decreased with the increase in the duration of cell desiccation. The addition of glucose revealed differences between the studied strains. Glucose stimulated the respiration of *S. cerevisiae* 77 cells even after 22 h of desiccation. Strain 14 cells suspended respiration earlier than strain 77 ones during dehydration and, after 14 h of drying, stopped reacting to the addition of glucose.

The studied cells differed not only by the accumulated amount of K^+^, but also in the ability to keep these ions inside the cells during dehydration. Intensive K^+^ leakage from strain 77 cells was observed till the sixth hour of desiccation and later only a very slight K^+^ flow was observed. Strain 14 cells gradually released intracellular K^+^ during the first 14 h of desiccation. After the addition of the antifungal polyene macrolide nystatin to the medium, the maximal amount of K^+^ leaked from the cells. The studied strains differed in their sensitivity to this pores in the yeast membranes forming compound. In suspensions of strain 77 cells, inhibition of respiration and nystatin-induced release of K^+^ started almost immediately after the addition of this antifungal. In the case of strain 14 cells, a 3–4 min lag period was observed after the addition of this compound and the effect of nystatin on the respiration activity of these cells was considerably weaker compared to strain 77. It is worth mentioning that in the case of strain 77 cells, nystatin released some amount of accumulated K^+^ even after 20 h of desiccation, although all accumulated K^+^ leaked out of strain 14 cells in 16 h.

Lipid bilayers bind rather large amounts of lipophilic anions [11]. PCB^−^ accumulates in yeast membranes after metabolic inactivation of the cells [12,13]. During dehydration, the binding of PCB^−^ indicated considerably higher metabolic activity of *S. cerevisiae* 77 cells compared to strain 14. The maximal amount of PCB^−^ was bound to strain 14 cells after 12 h of dehydration. Accumulation of this lipophilic anion in strain 77 cells gradually increased during 22 h of desiccation and nystatin induced an additional binding of this indicator to the cellular membranes (Figure 1E,F).

The different rates of changes in energetic characteristics of the studied strains could be a result of different rates of desiccation. Control experiments showed that before desiccation, strain 77 cells contained ~4% less water than strain 14 ones, but were losing it at a similar rate. Cells of both strains lost 70–80 % of their water during the first 11 h of desiccation (Appendix A).

### 3.2. Effects of Dehydration on Acidification of the Medium

The rate of acidification of the extracellular medium is frequently used to assay the rate of glycolysis, although there is another potential source of extracellular protons—respiration [14]. Monitoring the dissolved oxygen concentration in the medium demonstrated different effects of glucose on the respiration of the studied cells. Before drying, strain 77 cells actively reacted to the addition of glucose. Strain 14 ones actively consumed the dissolved oxygen before the addition of glucose, but their reaction to this substrate was less pronounced (Figure 1A,B).

The results of the medium acidification measurements showed (Figure 2) that, when responding to the addition of glucose before desiccation, *S. cerevisiae* 14 cells acidified the medium significantly more strongly compared to strain 77 ones. After 10 h of desiccation, the intensity of acidification in the case of *S. cerevisiae* 77 cells decreased almost four times and that of strain 14 cells around five times compared to the initial values before drying. It should be mentioned that strain 14 cells acidified the medium more than twice as efficiently. These results indicate that in experimental conditions, glycolysis is the main cause of medium acidification by yeast cells.

### 3.3. Efficiency of Rehydration

In the following experiments, the effects of rehydration (preincubation in deionized water) or reactivation (preincubation in YPD growth medium) on the energetics of the desiccated cells were analyzed. Rehydration slightly stimulated respiration of the dry *S. cerevisiae* cells and the effect on strain 77 cells was stronger compared to strain 14 ones (Figure 3A,B). The respiration activities of both strains were stimulated by the addition of glucose; in equilibrium conditions, the dissolved oxygen concentration in the *S. cerevisiae* 14 cell suspension was 83–84%, and ~78% in the case of strain 77 cells. The addition of nystatin induced a weak inhibition of respiration in the case of 77 cells only. Strain 14 cells preincubated in the deionized water for 30–180 min did not consume any dissolved oxygen. In the case of strain 77, preincubation of dried cells in the deionized water also gradually inhibited consumption of the dissolved oxygen, and after 180 min of preincubation, the oxygen consumption was close to 0%.

The accumulation of PCB^−^ ions indicates the ability of cells to repair their plasma membrane; only damaged cells bind large amounts of this indicator [10,13]. The desiccated cells directly added to the medium accumulated the largest amount of PCB^−^. The amount of the indicator bound to cells slightly decreased after preincubation of the desiccated cells in the deionized water (Figure 3E,F). Cells of both strains accumulated rather similar amounts of PCB^−^. Dependence of the amount of PCB^−^ bound on the duration of preincubation was a bit stronger in strain 14 cells, but strain 77 ones accumulated a larger amount of PCB^−^ after the addition of nystatin.

Measurements of K^+^ in the suspensions of dry cells resuspended in the deionized water showed that during desiccation, most of this ion leaked out of the cells. Nystatin released almost equal amounts of K^+^ from strain 77 cells independently of the duration of preincubation in deionized water. In the case of strain 14 cells, some nystatin-dependent release of K^+^ was observed only in the case of non-preincubated cells. After the preincubation of these cells in the deionized water, nystatin was not able to release any K^+^ from the cells. The amount of K^+^ in the medium indicated that most of the preincubated strain 14 cells lost their ability to accumulate this ion.

In summary, the results of our experiments indicated that after preincubation in the deionized water, strain 14 cells lost their respiration activity and ability to accumulate K^+^. However, strain 77 cells after preincubation in the deionized water demonstrated some respiration activity, maintained the nystatin-sensitive barrier to PCB^−^, and were able to accumulate K^+^, if the preincubation did not continue longer than 2.5 h.

During reactivation in the YPD growth medium, the cells of both strains restored their physiological activity more efficiently compared to rehydration in deionized water (Figure 4). Strain 77 cells after 120–180 min of reactivation and the addition of glucose very strongly consumed the dissolved oxygen. In equilibrium conditions, the concentration of the dissolved oxygen was only ~40% of the initial level. However, strain 14 cells consumed the dissolved oxygen considerably more weakly and, in equilibrium conditions, the concentration of the dissolved oxygen in the medium decreased only to 85–80%. Such low activity levels of the respiration of these cells were observed regardless of the duration of the reactivation period.

Measurements of K^+^ (Figure 4C,D) also showed different capabilities of the strains to restore the gradient of this cation on the plasma membrane of cells. After addition to the incubation medium, cells of both strains slowly released K^+^, but the main amount of this cation was released after nystatin addition to the cell suspensions. Strain 77 cells released more K^+^ than strain 14 ones and the accumulated amount of this ion inside the cells was dependent on the duration of reactivation. It is worth mentioning that in both cases, the nystatin-induced release of the intracellular K^+^ proceeded after a considerable lag period.

Accumulation of PCB^−^ by strain 77 cells gradually decreased during reactivation (Figure 4E,F) and, after 30 min of preincubation in YPD medium, these cells bound considerably less of this indicator. However, preincubation in the growth medium had a considerably weaker effect on the strain 14 dry cells, as they bound a larger amount of PCB^−^ regardless of the duration of reactivation.

### 3.4. ATP Content of the Cells during Dehydration and Rehydration

Samples for luciferin–luciferase luminometry of ATP content were taken from the vessels for electrochemical measurements after the transfer of cells from tubes for drying into 100 mM sodium phosphate buffer and incubation for 10 min.

The initial amount of ATP in strain 77 cells was approximately 20% higher compared to strain 14 ones (Figure 5A). During the first 2.5 h of desiccation, the amount of ATP in yeast cells increased by 35% in the case of strain 77 and by ~15% in strain 14 cells. However, after 5 h of desiccation, the ATP content decreased in both cells, with it being ~15% higher in strain 77. After 10 h of drying, in the case of strain 14, and after 12 h in strain 77 cells, the intracellular ATP content dropped to less than 1% of the initial level and stayed rather stable during the next 8–10 h of desiccation. During the first 3 h of reactivation in the YPD growth medium, the intracellular ATP content of strain 77 cells slowly increased, but it slowly decreased in the case of strain 14 cells.

## 4. Discussion

The main goal of this study was to determine factors important for the survival of yeast cells during their transition into the state of anhydrobiosis and out of it, with the aim to analyze the possibility of them having highly active biological systems after reactivation of the dry cells. It is important to enrich our knowledge about the main biochemical mechanisms working during the process of transition. Such findings would be key to improving methods for yeast preservation and to understanding the mechanisms of anhydrobiosis. Considering that yeasts are an optimal model of eukaryotic cells and are widely used for various studies in the fields of molecular biology, translational medicine, and pharmaceutics, they have more general importance, including the treatment of pathological states of the human body.

During desiccation, yeast cells face mechanical, structural, and oxidative challenges, including intracellular crowding, plasma membrane lysis, and permeabilization. To survive these stresses, yeasts have developed many endogenous protective mechanisms, including a unique elastic cell wall, the accumulation of intracellular glycerol and trehalose, the induction of stress proteins, and biochemical antioxidative systems [2,15].

It is still not clear why strain 77 cells are more resistant to desiccation than strain 14 ones. Most probably, there is a complex of reasons. In addition, resistance to desiccation depends on the growth phase. For the experiments, we used cells from the stationary growth phase. Such cells show better adaptation to the environment [3,5]. Results of the monitoring of respiration activities indicated the differences in behavior of the studied *S. cerevisiae* cells during dehydration: very resistant to dehydration strain 77 cells maintained higher glucose-induced respiration activity than semi-resistant strain 14 ones. Responding to the addition of glucose, *S. cerevisiae* 14 cells not only demonstrated weaker respiration activity, but also carried out more intense acidification of the incubation medium.

It is known that energy-deprived yeasts acidify their cytoplasm. This feature is related to the dormancy state of cells [16,17]. The increase in acidity of the cytosol during dormancy causes many proteins to interact with each other and form large clumps or filament structures that result in the cytoplasm becoming stiffer. Mobility of the organelles decreases, and mechanical stability of the cells increases. The results of our experiments indicate that strain 14 cells acidify the medium almost four times more efficiently compared to strain 77 cells. After 10 h of desiccation, the medium acidification considerably decreased, but strain 14 cells still extruded acid twice as extensively. It is possible that a lower extrusion of acidic products, but more intense respiration (stronger extrusion of H^+^ from mitochondria to the cytosol), helps strain 77 cells to reach the dormancy state more efficiently than strain 14 cells do.

Our results revealed a decrease in the ATP content in cells of both strains during dehydration. This probably indicates the usage of ATP for metabolic processes in the cells for intracellular protection of the cell structure and the maintenance of viability. While losing water content, yeasts synthesize endogenous compounds, such as glycerol and trehalose [2]. A rapid change in the cellular energy metabolism reflects the higher energy demands required for surviving solute synthesis against osmotic shock [18,19]. After ATP depletion, Marini et al. [16] observed significant cytosolic compaction and extensive cytoplasmic reorganization, as well as the emergence of distinct membrane-bound and membraneless organelles. The levels of ATP in both cells decreased with similar kinetics, but very resistant to dehydration 77 cells contained 10–20% more ATP than semi-resistant strain 14 ones. The larger amount of ATP in strain 77 cells before dehydration can also support the higher resistance of these cells to desiccation.

During reactivation, strain 77 cells more efficiently recovered intracellular ATP than strain 14 ones; the ATP level in strain 77 cells slowly increased, while the ATP content of strain 14 cells decreased. These results support the idea about the higher stability and activity of mitochondria in strain 77 cells, giving them the ability of more efficient repair of their structure and functions. However, before decreasing, the amount of ATP increased by 35% in the case of strain 77 and by ~15% in strain 14 cells during the first 2.5 h of desiccation (Figure 5). The increase in the ATP level could be a result of a temporal stimulation of synthesis or a consequence of an abrupt ceasing of ATP-consuming processes inside the cells. In our mind, the latter process would be more probable. If desiccation initially decreases the ATP consumption, in conditions of good aeration, the yeast cells in thin pellets formed after centrifugation continue oxidative phosphorylation at a high rate.

The plasma membrane barrier to PCB^−^ (Figure 4) and ATP content of strain 77 cells (Figure 5B) grew rather slowly. This could be due to very active consumption of ATP by the processes of cell recovery. The efficiency of ATP hydrolysis depends on Mg^2+^ ions, and these cations from the YPD medium could make ATP bioavailable [18] for many antioxidative systems to abrogate the resulting damage [2]. This could be one of the reasons why the cell reactivation in the YPD medium is more efficient than rehydration in water.

Erkut and colleagues [19] studied entry into the state of anhydrobiosis of two eukaryotic organisms—*S. cerevisiae* and the roundworm *Caenorhabditis elegans.* Both organisms entered a state of anhydrobiosis during cellular survival under conditions of starvation. They discovered that a glyoxylate shunt is the metabolic switch in both organisms, which helps them to survive desiccation by enabling or promoting gluconeogenesis for the biosynthesis of trehalose. They compared the energetic/metabolic states and suggested that hypoaerobic lifestyle could be a feature of desiccation tolerance. In our case, mid-stationary phase *S. cerevisiae* cells were not starving and were studied in aerobic conditions. Strain 14 cells started to demonstrate damaged respiration at rather early stages of dehydration; after 14 h of drying, these cells ceased the consumption of oxygen and did not demonstrate any reaction to the addition of glucose. However, this substrate was able to energize strain 77 cells even after 22 h of dehydration (Figure 1B).

Intracellular K^+^ helps to keep the osmotic balance across the plasma membrane and stabilizes the turgor pressure. During the dehydration of strain 14 cells, the reduction of the K^+^ gradient on the plasma membrane proceeded more slowly and after 8 h of desiccation, these cells contained more K^+^ than strain 77 ones. On the other hand, the addition of nystatin revealed that during 8–20 h of desiccation, strain 77 cells were in a “low, but stable K^+^” state. Strain 14 cells were accumulating K^+^ till around the fourteenth hour of desiccation, but they did not get to such a state; starting from the 16th h, nystatin did not induce any leakage of accumulated K^+^. It is possible that nystatin released K^+^ accumulated in the mitochondria of strain 77 cells. The activity of K^+^ uptake systems in *Saccharomyces* (i.e., Trk uniporters) and/or the corresponding gene expression can vary among the strains, and the cells can have different membrane potential, which is a driving force for K^+^ uptake. After the reduction of the driving force, the release of accumulated K^+^ can occur via transporters of this ion in the plasma membrane [20,21]. Our results suggest that the different modes of accumulation or the ability to maintain the amount of intracellular K^+^ in *S. cerevisiae* cells could affect the osmotolerance of the cells and be one of the factors determining the higher viability of strain 77 cells after rehydration.

PCB^−^-binding measurements demonstrated an efficient barrier of the plasma membrane to lipophilic anions in strain 77 cells. Binding of PCB^−^ to desiccated strain 77 cells decreased gradually, but after 30 min of reactivation in the YPD medium, these cells bound considerably smaller amounts of this compound. Restoration of the barrier properties of semi-resistant strain 14 cells was less efficient, but the reactivation worked much better than rehydration in the deionized water. These results indicate that nutrients are needed for the active recovery of cells. As already mentioned, the remaining K^+^ in the cells could help strain 77 yeasts to resist hyperosmotic stress. K^+^ transporters in the plasma membrane depend on the stability of the electrical potential difference and integrity of this membrane [22]. The accumulation of ergosterol in the plasma membrane leads to a higher stability of S. cerevisiae thermotolerant strains during dehydration compared to mesophilic strains [8,23]. Our results support the idea that higher stability of the cell membranes in the presence of ergosterol could help yeasts to resist desiccation and it could be related to better maintenance of K^+^ in mitochondria.

Summarizing the results of our study, we suppose that at least two interconnected factors are responsible for the resistance of *S. cerevisiae* strain 17 cells to dehydration–rehydration: stability of mitochondria and intracellular content of ATP. The higher viability of *S. cerevisiae* 77 cells after reactivation could be linked to better maintenance of mitochondria in the dehydrated strain 77 cells and, correspondingly, to more efficient ATP synthesis by oxidative phosphorylation during reactivation. This means that the special stability of the mitochondria, improving biotechnological approaches or pretreatments of yeasts, could improve the resistance of cells to dehydration-rehydration. This hypothesis, of course, should be verified in future experiments.

## Figures and Tables

**Figure 1 microorganisms-09-00444-f001:**
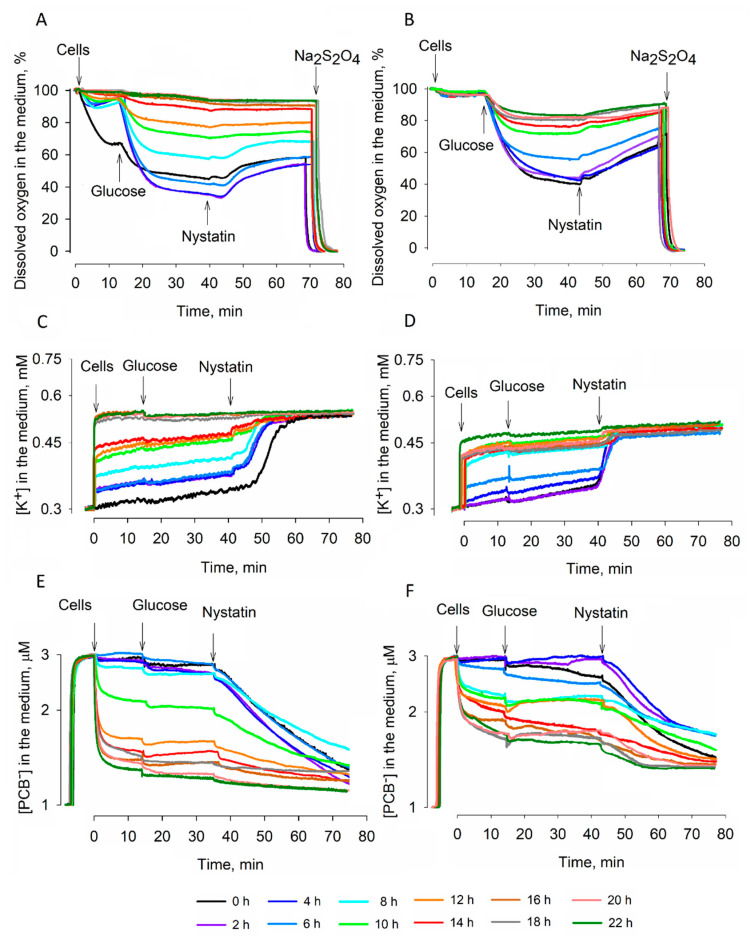
Changes in the energy status of *S. cerevisiae* 14 (**A**,**C**,**E**) and *S. cerevisiae* 77 (**B**,**D**,**F**) cells during dehydration. (**A**,**B**): activity of respiration, (**C**,**D**): ability to accumulate K^+^, (**E**,**F**): binding of phenyldicarbaundecaborane (PCB^−^) to the cells. Measurements were performed using magnetic stirring in thermostatted glass vessels. The cells after certain periods of desiccation were resuspended in 50 µL of 0.1 M sodium phosphate, pH 7.0, and added to 5 mL of the same buffer in the vessels. Glucose was added to a final concentration of 0.8% and nystatin to 10 µg/mL, respectively. To monitor dissolved oxygen, dry Na_2_S_2_O_4_ was added to a final concentration of ~20 mM.

**Figure 2 microorganisms-09-00444-f002:**
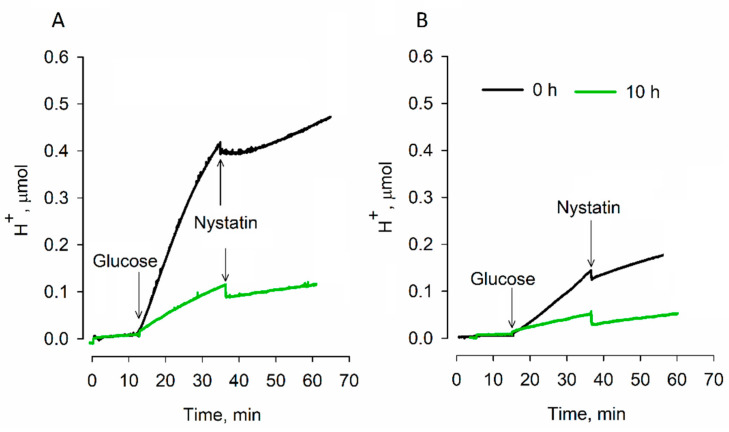
Effect of desiccation on the acidification of *S. cerevisiae* 14 (**A**) and 77 (**B**) cell suspensions. Measurements were performed at 30 °C in thermostatted and magnetically stirred glass vessels, containing 5 mL of 95 mM NaCl solution, buffered with 5 mM sodium phosphate, pH 7. Cells were analyzed before dehydration and after 10 h of drying. Glucose and nystatin were added to final concentrations of 0.8% and 10 µg/mL, respectively.

**Figure 3 microorganisms-09-00444-f003:**
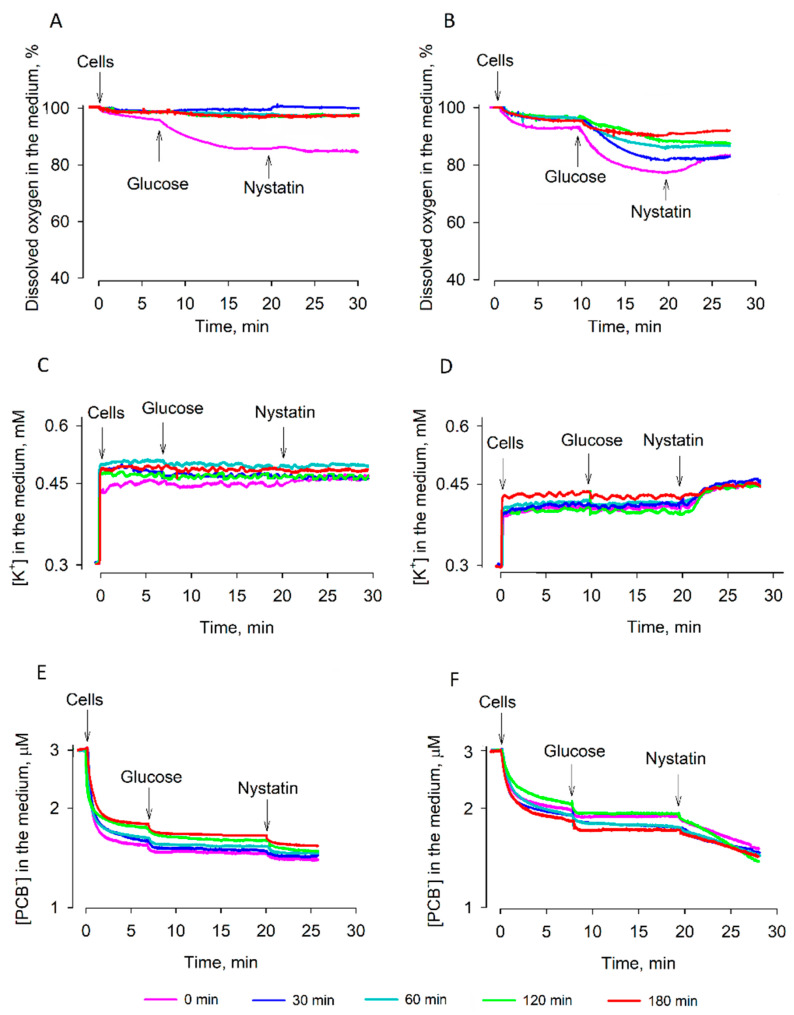
Changes in the energy status of dry *S. cerevisiae* 14 (**A**,**C**,**E**) and *S. cerevisiae* 77 (**B**,**D**,**F**) cells during rehydration in deionized water. (**A**,**B**): Activity of the respiration, (**C**,**D**): ability to accumulate K^+^, (**E**,**F**): binding of PCB^−^. The dry cells were rehydrated at room temperature by placing them for a definite period into deionized water. After rehydration, 3.4 × 10^8^ pelleted cells were added to 5 mL of 0.1 M sodium phosphate, pH 7.0. The measurements were performed in thermostatted glass vessels with magnetic stirring. Glucose was added to a final concentration of 0.8% and nystatin to 10 µg/mL. To monitor the dissolved oxygen, dry Na_2_S_2_O_4_ was added to a final concentration of ~20 mM.

**Figure 4 microorganisms-09-00444-f004:**
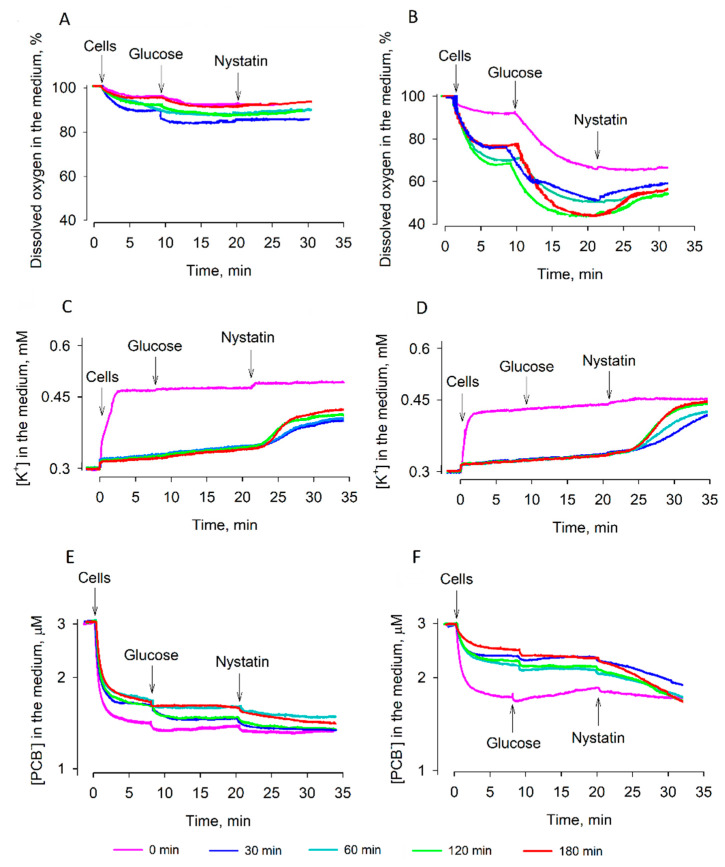
Changes in the energy status of dry *S. cerevisiae* 14 (**A**,**C**,**E**) and *S. cerevisiae* 77 (**B**,**D**,**F**) cells during reactivation in YPD medium. (**A**,**B**): activity of respiration, (**C**,**D**): ability to accumulate K^+^, (**E**,**F**): binding of PCB^−^. Dry cells were reactivated at room temperature by placing them for a definite period in YPD growth medium. After reactivation, 3.4 × 10^8^ pelleted cells were added to 5 mL of 0.1 M sodium phosphate, pH 7.0. Measurements were performed in magnetically stirred and thermostatted glass vessels. For K^+^ measurements, the reactivated cells were washed twice with 0.1 M sodium phosphate, pH 7.0. Glucose was added to a final concentration of 0.8% and nystatin to 10 µg/mL. To monitor the dissolved oxygen concentration, dry Na_2_S_2_O_4_ was added to a final concentration of ~20 mM.

**Figure 5 microorganisms-09-00444-f005:**
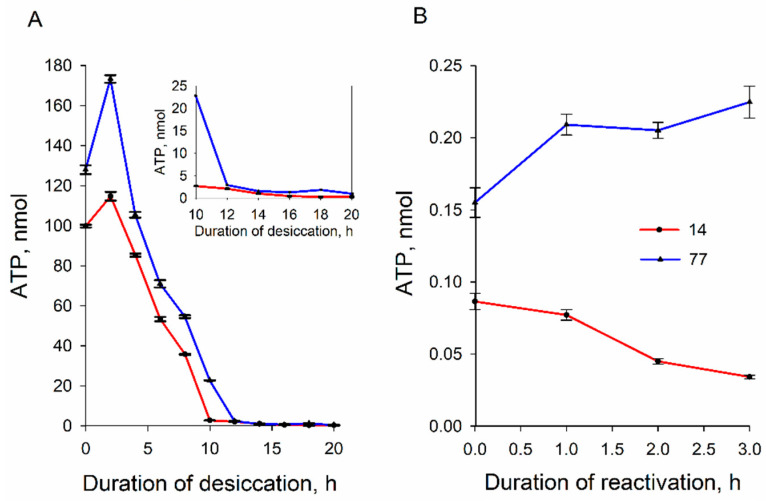
Changes in ATP content of *S. cerevisiae* 14 and 77 cells during their desiccation (**A**) and reactivation in the YPD medium (**B**). The amount of ATP was determined using a luciferin–luciferase bioluminescence assay, taking 5 μL glucose-containing samples (3.4 × 10^5^ cells) from vessels. Data are presented as mean ± S.D. (*n* = 3, three independent experiments).

## Data Availability

Data available in a publicly accessible repository that does not issue DOIs.

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
