# Peer review of "Changes in Energy Status of Saccharomyces cerevisiae Cells during Dehydration and Rehydration"

_microorganisms, 2021, doi:10.3390/microorganisms9020444_

Round 1

Reviewer 1 Report

The authors responded to most of the reviewers' comments and recommendations. The revision clearly contributed to the overall quality of the manuscript. 

Reviewer 2 Report

The authors took into account the remarks of the different referees and many corrections were performed. The new version of the manuscript is really better than the previous one.

This manuscript is a resubmission of an earlier submission. The following is a list of the peer review reports and author responses from that submission.

Round 1

Reviewer 1 Report

Summary

Yeast occupy a wide range of environmental niches and have differences in tolerances to different stressors, including dehydration. Yeast in industrial applications are often dehydrated and the most useful ones become metabolic active soon after rehydration with reduced membrane leakage. In this study, five different compounds from two divergent yeast strains were measured after different periods of dehydration and then rehydration in different conditions. The decrease of dissolved oxygen was used as a proxy for respiration. Increased potassium indicated leaky cellular membranes. Phenyldicaraundecaborane is a lipophilic anion. The acidification of the media and ATP levels were also measured. Rehydration in YPD appeared to reduce membrane leakage and decreased the time needed to return to be metabolic active only in the desiccation-tolerant yeast compared to water.

Comments

1. There needs to be more characterization on the two strains, 14 and 77. The link to the website does not provide any information on the yeast. Where were they isolated from? What else is known about them? Do they have different morphologies?

2. How is desiccation tolerance determined? Did both cells have the same amount of water before drying? How was the condition for desiccation determined? How did the authors ensure that the strains were completely dried out before rehydration? Were the yeast dried in the same way as if they would be for industrial applications? If the eventual application is the production of dried yeast, then the return to normal growth needs to be quantitated by either growth curves or colony-forming units. The metabolic measurements were conducted only over a short period of time and long-term outcomes were not measured.

3. With only two strains in the study, measurements of dissolved oxygen, potassium, and PCB in the media and without further analysis appear to be correlation and causation hasn’t been determined. The levels of these three molecules were not manipulated to determine if they would affect growth after rehydration.

4. How different are 14 and 77 in regular growth conditions? At times authors mentioned that 77 was thermotolerant. Is that something that correlates with desiccation tolerance? Is strain 77 just hardier than 14? How about other stressors like salt, extreme pH, DTT, cold, alternative carbon sources?

5. There is no information in the introduction as to why phenyldicarbaundecaborane was measured as cells were rehydrated. What does this compound do in yeast? Is the biosynthetic pathway known? If this manuscript aims to determine differences between yeast that would lead to the differences in recovery from dehydration, then there are better-characterized biomarkers such as trehalose and hydrophilins. Also, there was no mention of the cell wall and its components such as chitin, glucans, etc or how the cell wall could affect survival. On line 50, the authors state, “In order to be essential an experiment needs to be done that removes said chemical and then adhydrobiosis is inhbited.” However, PCB was not removed or altered and so the role in rehydration has not been shown to have a role in this process.

6. From the methods, it appears that there were biological replicates in the metabolic measurements but there no error bars or statistical analyses done on any data presented. How variable were the measurements? Are the differences between the strains statistically significant?

7. Wouldn’t the measurements of the ATP/ADP be a better measurement of the energy status in the cell? The drop in oxygen, increase in potassium and PCB are indirect measurements.

8. More information on the electrochemical monitoring system mention on line 99 is needed because this is central for understanding the experiments.

10. In figure 1A and 1B, how do these results compare to strain 14 and 77 grown in YPD in log-phase? Do desiccation resistant cells use less oxygen in log-phase? Are they doing more glycolysis during log-phase or recovery? Does strain 77 use non-fermentable carbon sources differently than strain14?

11. In figure 1C and 1D, as cells are rehydrated more potassium leaks out into the media from the dehydration sensitive cells. Do these cells have more potassium, to begin with?

12. In figure 1E and 1F, the sensitive cells release less PCB into the media when they have been dehydrated for longer periods of time compared to the resistant cells. It seems that 10 hours is the only intermediate condition. Is the 10-hour time point statistically different from the 0-8 hour time points and 12-24 hour time points?

14. In figure 2, it appears that the acidification of the media had not plateaued. Would the accumulation of H+ become equal between the two strains given enough time?

15. In Figure 3, is the addition of glucose, in the YPD or a separate addition of glucose? More explanation about how Figure 1 is different from Figure 3. How long were cells dried before rehydration? Were the cells all dried for the same amount of time and then they were rehydrated for different periods of time before the measurements were made?

16. In Figure 4, it appears that the cells were rehydrated in YPD that already contains 2% glucose. How would the addition of 0.8% more glucose affect cells?

17. In figure 5A, ATP levels of strain 77 are already higher than strain 14. Does strain 77 grow faster in log-phase? Is there a hypothesis as to why ATP levels appear to briefly increase at 2.5 hours of drying? Are the levels of ATP at 20 hours in Figure 5A at the same levels as 0 hours in 5B?

18. Supplemental figure 1 measures the change in weight after cells were dried for several timepoints but there are more time points presented in the figures. It assumes that only water is evaporating. There are other volatile compounds. There are error bars on this graph and while there are no statistics, the 10-hour and 24-hour time point looks different. How do the authors know that the weight of the pellet does not further decrease after 24 hours of drying?

19. Does starvation of yeast before dehydration improve recovery?

20. The discussion is largely a summary of the results and doesn’t put the results in the context in the field other than talking about Trk uniporters. Is anything known about these proteins in these two strains?

21. There are several instances where grammar and spelling should be improved for clear meaning. There are several free online grammar programs such as Grammarly that have plugins for Word.

Reviewer 2 Report

This manuscript is interresting but very difficult to read and to understand.

Abstract : do not write 77 strain cells and 14 cells without describing what it means. except you, nobody knows 77 and 14 strains. Please rewrite the abstract.

line 47 : no s at yeast

line 49 add a to between effectively and transit

line 75 : use passive form

line78 : did you check that you reached the mid of the stationnary phase? How?

line 82 : add wavelenght

line 90 : reformulate the sentence

line 106 : 0.8% P/V%

lines 113  and 114 : write were instaed of was

please add if you performed statistical analyses

Results part :

This part is not clear at all and must be rewriten. There are too many results presented only with graph.

You could maybe suppress experimental values on your graphs, especially when the observed phenomenom is the same. you ested every 2 hours. maybe with the results of every 4 hours, the graphs would be easier to read.

Insert tables to compare the results in which, you could insert rates (as presented line 157), speed etc.....

The reading will be easier.

I propose the same for the discussion part.

I propose you to reformulate and rewrite the manuscript before a new evaluation.

Reviewer 3 Report

In this manuscript, Kuliesiene and co-authors present a study on the changes in the energy status of Saccharomyces cerevisiae cells during dehydration and rehydration processes. The topic of the investigation is interesting, as strains resistant to environmental stresses such as temperature shocks, high temperature variations or dehydration/rehydration are prerequisites for a great number of biotechnological applications.

The manuscript is generally well conceived, presenting clear and well-structured data. There are some issues which the authors need to consider though, before this manuscript could be considered.

  • The main concern of this reviewer is the insufficient information provided concerning the yeast strains utilized in this study. The authors merely specify that the strains used were the „very-resistant to desiccation strain 77” and „mesophilic strain 14”. What is the genetic background of the strains? Are they laboratory or industrial strains? Are the strains used isogenic?  This latter aspect is crucial for the study, otherwise the comparison between the two strains is not substantiated.
  • Better description of the strains 77 and 14 is necessary in the Materials and Methods section The strain source is specified, but following the link provided by the authors (page 2, line 75) not much information can be accessed.
  • The authors may consider re-naming the two strains used in the study and use italics for these names, for an easier delimitation within the text.
  • The abstract is a bit confusing (mainly because of the strain denomination). For example, when first reading the abstract, this reviewer misunderstood „the respiration activity of very resistant to desiccation 77 cells” for 77 different cells (hence 77 different strains!, see lines 20-22). Same confusion in lines 131-132, 127-129, etc.